# Rice SUT and SWEET Transporters

**DOI:** 10.3390/ijms222011198

**Published:** 2021-10-18

**Authors:** Zhi Hu, Zhenjia Tang, Yanming Zhang, Liping Niu, Fang Yang, Dechun Zhang, Yibing Hu

**Affiliations:** 1College of Resources & Environmental Sciences, Nanjing Agricultural University, Nanjing 210095, China; 2019203047@njau.edu.cn (Z.H.); 2019103102@njau.edu.cn (Z.T.); 2State Key Laboratory of Hybrid Rice, College of Life Sciences, Wuhan University, Wuhan 430072, China; sklhr@whu.edu.cn (Y.Z.); lipingniu@whu.edu.cn (L.N.); fang-yang@whu.edu.cn (F.Y.); 3Bio-Technology Research Center, China Three Gorges University, Yichang 443002, China

**Keywords:** rice, SUT(C), SWEET, sugar, transporter

## Abstract

Sugar transporters play important or even indispensable roles in sugar translocation among adjacent cells in the plant. They are mainly composed of sucrose–proton symporter SUT family members and SWEET family members. In rice, 5 and 21 members are identified in these transporter families, and some of their physiological functions have been characterized on the basis of gene knockout or knockdown strategies. Existing evidence shows that most SUT members play indispensable roles, while many SWEET members are seemingly not so critical in plant growth and development regarding whether their mutants display an aberrant phenotype or not. Generally, the expressions of *SUT* and *SWEET* genes focus on the leaf, stem, and grain that represent the source, transport, and sink organs where carbohydrate production, allocation, and storage take place. Rice SUT and SWEET also play roles in both biotic and abiotic stress responses in addition to plant growth and development. At present, these sugar transporter gene regulation mechanisms are largely unclear. In this review, we compare the expressional profiles of these sugar transporter genes on the basis of chip data and elaborate their research advances. Some suggestions concerning future investigation are also proposed.

## 1. Introduction

Sugar is synthesized in the green tissues of plants and is transported into the sink organs to be stored as starch or other organic compounds, in addition to being metabolized as an energy-producing molecule. Translocation of sugar plays an indispensable role in plant growth and development through controlling carbon partitioning between the stem and grain [1]. Sucrose is the main form of carbohydrates for long-distance transport in plants. It is mainly translocated via the symplastic pathway composed of sieve elements–companion cells (SE–CC complexes) in the vascular tissues from leaf to flower, seed, and root in higher plants [2,3]. For short-distance transport of sugar, usually between adjacent cells, the apoplastic pathway that depends on membrane-located transporters plays a critical role [4,5]. Due to two outstanding discoveries of sugar transporters, namely, SUT(C) (sucrose transporter or carrier) and SWEET (sugar will eventually be exported transporter), in plants achieved in the past 30 years [6,7], the molecular mechanism of sugar apoplastic transport in model plants is beginning to be clarified.

SUT is a class of sucrose–proton symporter only present in plants, probably because sucrose is not produced in animals and microorganisms. As a group of positive sugar transporters, they are extremely important for the acquisition of sucrose from the intercellular apoplast outside of the cell into the cytosol, particularly when no plasmodesma exists between adjacent cells, and a membrane transporter or channel is the only pathway for material exchange. These circumstances include the maternal–filial interface in the seed, the embryo–endosperm interface after the initial stage of seed development, and probably the pollen tube–pistil interface during the pollen tube fast growth stage after pollination [8,9,10]. Moreover, in long-distance transport of sugar via the symplastic pathway, sucrose must be first uploaded into the phloem. The uploading process relies on SUT transporter(s) since very few plasmodesmata exist between the SE–CC complexes and their surrounding cells [11,12]. Additionally, during symplastic transport within the phloem, a small portion of sucrose may leak into the apoplast outside of the SE–CC complexes; retrieving this portion of sucrose back to phloem also requires SUT transporter’s participation [13,14].

Single-cell organisms only need to assimilate sugar from their environment or produce sugar by themselves if they live on it. However, multicellular organisms also need to export sugar from one cell to another for sugar allocation between cells. Moreover, the efflux of sugar from plant cells is necessary for the symbiont or pathogen’s living [15]. In addition, glucose, as the most important molecule for cell metabolism, is universally used from bacteria and archaea to plants and animals. SUT transporters, however, are unable to undertake the transport of glucose due to a lack of substrate compatibility [16]. The identification of *SWEET* genes revealed how the needs listed above were met in the cell. 

In 2010, Chen et al. [7] characterized the first *SWEET* gene, *AtSWEET1*, in *Arabidopsis*. SWEET is a class of passive sugar transporters that transport oligosaccharides such as glucose or sucrose across the membrane along their concentration gradients. This property attributes them with the ability to import or export sugar in or out of a cell. Unlike the plant-specific SUT, SWEETs are present in both plants and animals [17]. Moreover, their homologs in prokaryotes, SemiSWEETs, which also transport sucrose and glucose, were identified in both bacteria and archaea [18]. Recently, a membrane protein of SARS-CoV-2 (QJA17755) was postulated to resemble SemiSWEETs, as the putative protein possesses a triple helix bundle and forms a single three-transmembrane domain [19]. However, sequence alignment showed that the amino-acid identity of the protein with BjSemiSWEET1 [18] and EcSemiSWEET [20] is only about 15.32%, and no MtN3/slv domain can be identified in the membrane protein of SARS-CoV-2 (http://pfam.xfam.org/search/sequence, accessed on 5 October 2021). By contrast, a putative SemiSWEET (DAE96463) of *Myoviridae* sp. (phage) from Human Metagenome was retrieved by similarity search. It contains a MtN3/slv domain and shows 31.93% amino-acid identity with BjSemiSWEET1 and EcSemiSWEET. This result indicates that *SWEET* homologs are distributed much more widely than *SUT* genes. Given that no SemiSWEET functions have been physiologically characterized to date, a virus SemiSWEET, e.g., DAE96463, may be a new choice for a breakthrough in this area.

In this review, we focus on the two classes of sugar transporters in rice and review advances in the characterization of their physiological roles and molecular mechanisms. Gaining a full understanding of the functions of these sugar transporters is challenging, particularly with respect to gene regulation mechanisms.

## 2. Physiological Functions of Rice SUT Sucrose Transporters

*SUT*s are intensively investigated in model plants despite the first *SUT* gene being characterized in Spinach [6]. In rice, only five transporter genes have been identified in the *SUT* family [21], but much attention has been paid to their physiological functions, particularly *OsSUT1*. Since its first report in 1997 [22], many roles of the transporter have been documented via Tos17 or T-DNA insertion-mediated heterologous mutants or RNAi/antisense-mediated knockdown lines of *OsSUT1*. These functions include providing sugar for pollen development, pollen germination and seed germination [23,24], uploading sucrose into the phloem for long-distance transport [13,25,26], and retrieving leaked sucrose from apoplast outside of the SE–CC complex during sucrose long-distance transport [13]. The most important role it plays is probably in seed-filling [27,28,29,30,31,32,33], since the CRISPR/Cas9-mediated mutation of the gene conferred complete infertility although the mutant plants did not show much difference from the WT control at their vegetative growth stage except for a slightly dwarfed size [34]. This result is generally consistent with the observations that a homozygous *ossut1* mutant derived from anther tissue culture [35] and transgenic rice lines with antisense *OsSUT1* expression did not confer any abnormal phenotype at the vegetative growth stage [28,36].

In higher plants, phloem loading of sucrose can be accomplished either via the apoplastic pathway which depends on membrane-located sucrose transporters or via the symplastic pathway which depends on plasmodesmata between SE–CC complexes and adjacent cells [3,37]. These pathways are usually present in herbaceous plants including crops and trees [14,38]. OsSUT1′s sucrose-transport capacity [16,21], localization on the PM of companion cells [34,39], and ability to complement *atsut2* [25] support the apoplastic pathway of phloem loading in rice. However, the symplastic pathway of phloem loading was also proposed previously by Kaneko et al. [40] through their observation of the ultrastructure of small vascular bundles in rice seedlings. Considering that knockout or knockdown of *OsSUT1* did not impair rice vegetative growth [28,35,36], and other SUT members of rice either belong to different types or have a different expressional pattern from OsSUT1 [41,42], Eom et al. [35] proposed that the apoplastic pathway might not be the primary route for rice phloem loading. Therefore, whether the apoplastic or the symplastic is predominant in rice phloem loading remains an open question [14].

Under normal growth conditions in the paddy field, however, *ossut1* homozygous mutants were infertile due to failed grain-filling [34]. This suggests that the failed reproductive growth in the mutants was resulted from the dysfunction of apoplastic phloem loading undertaken by OsSUT1. This pending issue can probably be reconciled from a new perspective: the apoplastic transport of sucrose assumed by OsSUT1 in the SE–CC complexes is dispensable when the amount of sucrose for phloem loading is minimal at the vegetative growth stage of rice; however, it may be critical when a large amount of sucrose needs to be uploaded into the SE–CC complexes for long-distance transport into grains during the reproductive growth stage.

Compared with OsSUT1, the remaining members of the rice SUT family seem to be less critical since mutants of their encoding genes are generally fertile. However, the production and the growth of these mutants are impaired to different extents (Table 1). OsSUT2 was identified to localize on cell tonoplast; it participates in plant growth, and knockout of *OsSUT2* reduced tiller number, plant height, grain weight, and root dry weight of rice [43,44,45]. OsSUT3 is a protein specifically expressed in pollen, and it may be important for pollen and starch accumulation [46,47]. The plasma membrane-localized OsSUT4 is expressed in the vascular tissue of the embryo and coleoptile in the germinated seed; it is also expressed in glume shells, anthers at the flowering stage, and the aleurone layer of caryopsis at the seed-filling stage [48,49,50]. Knockout of *OsSUT4* dwarfed the mutant rice lines, as well as reduced their tiller number and grain yield [48,49,50]. OsSUT5 is expressed in the culm, leaf, inflorescence, and caryopsis of rice, but it is predominantly expressed during inflorescence and caryopsis development at the transcriptional level. OsSUT5 was identified to localize on the plasma membrane; knockout of *OsSUT5* reduced the rice seed-setting rate and conferred a chalky endosperm of the mutant caryopses [51,52].

A comparison of rice *SUT* gene expressions based on chip data shows that their expressional intensities seem to match the physiological roles played by their gene-encoding proteins (Table 2). For example, OsSUT1 is critical for rice development at the reproductive growth stage, and *OsSUT1* shows the strongest expression among rice *SUT* genes, particularly in the stem and the developing caryopsis. The expression of *OsSUT2* is weaker than that of *OsSUT1* but stronger than that of the remaining *SUT* genes of rice. Accordingly, the abnormality of *OsSUT2* mutants is milder than that of *OsSUT1* mutants but greater than that of the remaining rice *SUT* gene knockout mutants. Moreover, the expressions of rice *SUT* genes, especially *OsSUT1* and *OsSUT2*, focus on the leaf, stem, and grain (Table 2). These organs are the source, transport, and sink organs of rice, respectively. Notably, all *SUT* genes except for *OsSUT2* show very weak expressions in the root at the transcriptional level. Xu et al. [53] reported that none of the rice *SUT* genes were expressed in the root, as detected via GUS expression. However, another investigation employing the same method showed that OsSUT1 was expressed in the root of rice shortly after seed germination [34]. Different promoter lengths used for driving GUS expression or different observation times perhaps gave rise to these differences.

Interestingly, *Arabidopsis* possesses nine SUC members, while the rice SUT family contains five SUT members, although the *Arabidopsis* genome is only about one-third of the rice genome [21,41,42]. However, both *Arabidopsis* and rice contain 12 *SUT* gene transcripts according to a recent investigation [34]. Notably, *OsSUT1* possesses at least six alternative splicings of transcripts [34]. As alternative splicing is a way to regulate plant development [54], it probably explains why rice SUTs can assume so many roles in plant growth and development with limited members. Nevertheless, functional characterizations of all transcripts of *OsSUT1* and elucidating their regulation mechanisms remain challenging.

## 3. Rice *SUT* Gene Regulations

Not surprisingly, the expression of rice *SUT* genes is strictly regulated by various factors in changing environments. In 2012, Siahpoosh [55] reported that modification of *OsSUT1* expression regulated the salt response of rice *Oryza sativa* cv. Taipei 309. This suggests that the sugar transporter gene may participate in the cell’s abiotic stress response by adjusting cytosol sugar concentration. Under drought stress, the expression of *OsSUT1* was intensified and the expression of *OsSUT4* was downregulated significantly [56]. High temperature or heat stress during the seed-filling stage reduced *OsSUT1* and starch synthesis-related gene expression and led to earlier ripening and chalky grain [31,33,56]. On the other hand, low temperature or chilling treatment also downregulated the expressions of *OsSUT1*, *OsSUT2*, and *OsSUT4* [50,57]. In addition, the expression of *OsSUT1* was reduced in the leaf and grain but increased in the stem under submergence at the ripening stage of rice development [58]. 

It is easy to understand that CO_2_ concentration in the environment can affect the sucrose transporter gene’s expression [56]. However, the rice SUT gene expressions are also influenced by plant nutritional factors. For example, Chen et al. [59] reported that the deficiency of potassium induced by knockout of *OsHAK1* reduced the expressions of *OsSUT1*, *OsSUT2*, *OsSUT4*, and *OsSUT5* at both the vegetative and the reproductive growth stages of rice. Moreover, iron deficiency inhibited all *SUT* gene expressions in rice leaf [60]. Mutation of *PHO3* (*AtSUT2*) in *Arabidopsis* brought the mutant plant a phenotype of low phosphorus stress [61,62]. It is worth waiting to identify the corresponding *SUT* gene in rice. Nevertheless, this suggests that the plant phosphorus pathway and sucrose transport activity are interlinked. Nitrogen uptake is highly integrated with the availability of sugars [63]. However, it is still not clear whether the rice SUT gene expressions respond to nitrogen form or abundance, although a similar occurrence was identified in crabapple [64].

Interestingly, biotic stress such as insect infections may also affect *SUT* gene expression. Ibraheem et al. [65] reported that aphid feeding on rice leaf blade vascular bundles caused an upregulation of *OsSUT1* expression in xylem parenchyma. In addition, Chang et al. [66] reported that the larval infestation of *Cnaphalocrocis medinalis* and mechanical wounding induced an upregulation of *OsSUT4* expression. If these phenomena can be confirmed, whether this upregulation of *SUT* gene expression is a response to the drop in sucrose concentration in plant cells caused by insect feeding or the aftermath of the insect’s released substance can be investigated. In any case, this process helps the insects acquire more sucrose from the vascular tissue of rice they infected.

Despite that rice *SUT* gene expressions being influenced by a plethora of environmental factors including biotic and abiotic stresses, the underlying mechanisms are still unclear. In all of the responses of *SUT* gene expression to various environmental stimuli mentioned above, no transcription factor which directly binds to the promoter sequence of the *SUT* genes has been identified to date. As a result, no regulation pathway of these *SUT* genes can be established on the basis of current knowledge. By contrast, concerning the SUT protein’s role in rice growth and development, using a yeast one-hybrid assay and EMSA analysis, Bai et al. [67] demonstrated that *OsSUT1*, *3*, and *4* were regulated by a nuclear factor NF-YB1 during rice grain-filling. Knockout of the NF-YB1-encoding gene in rice conferred a chalky endosperm phenotype. This suggests that NF-YB1 works upstream of the *SUT* genes and is necessary for rice normal grain-filling. Moreover, a recent investigation showed that both *OsSUT1* and *OsSWEET11*, *14* were targeted by OsDOF11, a DNA-binding protein that controlled their expressions [68].

## 4. Physiological Functions of Rice SWEETs

Animals usually contain only one or a few *SWEET* members in their genomes, while flowering plants generally possess 17–20 members on average, according to investigations of more than 30 plant species [69,70]. SWEETs participate in a variety of plant activities, including seed-filling, nectar secretion, pollen nutrition, and phloem loading and unloading [17,71,72]. In rice, 21 SWEET members have been identified [73]. The first member characterized with a physiological function in seed-filling is *OsSWEET4*. Knockout mutants of the gene bore almost complete empty caryopses, and a similar phenotype was observed in the gene’s homolog mutants of Maize [74]. Later, Ma et al. [75] and Yang et al. [76] reported that knockout of *OsSWEET11* impaired grain-filling of the gene’s mutants, which demonstrated the essential role of the gene encoding protein in seed-filling. Furthermore, Ma et al. [75] reported that the seed setting rate of the mutants was reduced. Yang et al. [76] reported that the maturation of the gene’s mutants was postponed, and double mutation of *OsSWEET11* and *OsSWEET15* conferred complete infertility (Table 3). A recent investigation [77] confirmed the results of Ma et al. [75] and Yang et al. [76]. Previously, both Chu et al. [78] and Yang et al. [79] reported that OsSWEET11 played a role in rice pollen development via RNAi analysis.

In contrast to the results that knockout of rice *SUT* genes usually confers aberrant phenotypes (Table 1), some *SWEET* genes of rice seem dispensable in plant growth and development. Accumulating data show that knockout of *OsSWEET14* in different rice varieties did not cause any abnormal phenotype [77,80,81,82]. Moreover, we mutated *OsSWEET1a*, *OsSWEET14*, and *OsSWEET5* via CRISPR/Cas9-mediated genome editing in Japonica rice Nipponbare but did not observe any obvious abnormal phenotype [83].

From a comparison of the phenotypes of single or double mutants of characterized rice *SWEET* genes, it seems that *OsSWEET4* is the most important among them for rice growth and development, because mutation of the gene seriously affected seed-filling [74]. A less critical gene is probably *OsSWEET11*, since knockout of the gene significantly affected rice grain-filling [75,76,77] and knockdown of the gene reduced pollen viability [78,79]. Double mutation of *OsSWEET11* and *OsSWEET15* in rice led to complete infertility, although knockout of *OsSWEET15* alone did not cause an abnormal phenotype according to Yang et al. [75]. By contrast, double mutants of *OsSWEET11* and *OsSWEET14* were fertile although they suffered a heavier impairment in seed-filling than the *OsSWEET11* single-gene mutants [77]. It seems that the importance of the characterized *SWEET* genes in rice growth and development conform to the following order: *OsSWEET4 > OsSWEET11 > OsSWEET15 > OsSWEET14*.

Chip data quantification, as shown in Table 4, indicates that *OsSWEET1a*, *OsSWEET4*, *OsSWEET11*, *OsSWEET13*, *OsSWEET14*, *OsSWEET15*, and *OsSWEET16* show relatively strong expressions, with *OsSWEET1a* being the strongest among them. Many *SWEET* genes of rice show weak, marginal, or no expression. Unlike the expression intensities of rice *SUT* genes, which were generally consistent with the impairments their mutants suffered, *OsSWEET1a* mutants did not show any aberrant phenotype [83]. By contrast, the knockout and knockdown of *OsSWEET4* and *OsSWEET11* in rice led to obvious abnormalities [74,75,76,77,78,79] despite their expressions being weaker than that of *OsSWEET1a*.

As both SUT and SWEET can transport sugar across the membranes, it is interesting to determine which will have the predominant role when the sugar concentration outside of a cell is higher than that of the cytosol. Both SUTs and SWEETs can achieve this function under these circumstances. Comparatively, SWEET might be the economic choice because it does not consume proton potential which needs energy input during the transport process. The protein that will be recruited to do the job probably depends on which gene’s expression is induced under the circumstances. As low-affinity and pH-independent sugar transporter-encoding genes [84], *SWEET*s are usually expressed strongly in rice tissues with abundant sugar to transport under normal growth conditions, as observed in Table 4, except for *OsSWEET1a*. Therefore, it is not surprising that SWEET proteins usually act as sugar efflux transporters.

## 5. SWEETs Interaction with Pathogens

In addition to the roles they play in plant growth and development, some members in clade III of the rice SWEET family are deeply involved in plant–pathogen interactions. Due to their dual property of being able to transport sugar in and out of a cell across the plasm membrane, three genes from clade III of the rice *SWEET* family are liable to be hijacked by pathogens such as *Xanthomonas oryzae* pv. *Oryzae* (*Xoo*) and *Rhizoctonia solani* for their propagation [85,86].

*Xoo* is a widespread vascular pathogen of rice bacterial blight that prevails in Asia and Africa [86]. It causes severe yield loss of rice and seriously endangers food security. By targeting effector-binding elements (EBEs) in the promoters of susceptible *SWEET* genes with transcription activator-like effectors (TALe), *Xoo* induces *SWEET* gene expression to obtain sugar from the host cell for its propagation [7]. Currently identified TALes of *Xoo* can be classified into six types: PthXo1, PthXo2, PthXo3, AvrXa7, Tal F, and Tal C. *Xoo*-susceptible *SWEET* genes contain one or more of these TALe targets, i.e., EBEs, in their promoter regions [86]. The EBEs in the promoter regions of *OsSWEET11* and *OsSWEET13* are targeted by PthXo1 and PthXo2, respectively [86,87,88,89]. However, *OsSWEET14* is targeted by one or more TALes including PthXo3, AvrXa7, Tal F, and Tal C [86,89,90,91,92,93,94,95].

The first case revealing the relationship between a *SWEET* gene and *Xoo* was *Xa13* (*OsSWEET11*), which is a *Xoo*-related resistance locus in the recessive *xa13* rice strain [79,96]. Later, it was identified that rice *xa13* recessive resistance to bacterial blight can be defeated by induction of the disease-susceptible gene *Os-11N3* (*OsSWEET14*) [88].

Since four of the six major TAL effector (TALe) types of *Xoo* strains from different geographic origins and genetic lineages target its promoter [86,89,90,91,92,93,94,95], *OsSWEET14* is the pivotal target gene of all characterized *Xoo* varieties from Africa and most varieties from Asia. Thus, many efforts to avoid *Xoo* infection have focused on the regulation of this gene’s expression. Recently, using CRISPR/Cas9-mediated genome editing based on an *OsSWEET11* resistant allele cultivar, Oliva et al. [86] demonstrated that mutation of the EBEs in the promoter regions of *OsSWEET13* and *OsSWEET14* conferred a broad-spectrum resistance of rice to *Xoo* variety infections. This work highlights the promising feature of this strategy in combating *Xoo* infections.

Naturally occurring *Xoo*-resistant rice varieties are often derived from spontaneous mutation of the EBEs in the susceptible gene’s promoters. Similar to *OsSWEET11* and *OsSWEET13*, *OsSWEET14* also has a *Xoo*-resistant allele [95]. These *Xoo-*resistant alleles are valuable germplasms for breeding. As mentioned above, at least five independent investigations revealed that knockout of *OsSWEET14* in different rice cultivars (Kitaake; Zhonghua 11; Nipponbare) did not impair rice growth and yield [77,80,81,82,83]. Future breeding efforts to obtain *Xoo*-resistant rice cultivars based on *OsSWEET14* mutants naturally occurring or edited in the coding region of the gene may be another choice, since the existing *Xoo*-resistant alleles are usually derived from EBE mutations of the susceptible genes. An additional advantage of the alternative is that even the adaption of *Xoo* TALes cannot control the target gene for sugar acquisition.

## 6. *SWEET*s Response to Abiotic Stress and Regulation

The response of *SWEET* genes to abiotic stress was first identified in *Arabidopsis* [97,98]. More detailed information about the role of *SWEET* genes in response to abiotic stress such as cold, high temperature, drought, and salinity in a variety of higher plants can be found in an excellent review by Breia et al. [70]. Very recently, Mathan et al. [99] reported that the expressions of *OsSWEET13* and *OsSWEET15* were intensified at both the transcriptional and the translational levels in response to salt and drought stresses. It will not be surprising if future investigations reveal that rice sugar transporter genes are involved in all kinds of sugar-associated activities.

The regulation mechanism of the expression of *SWEET* genes by pathogen TALes is relatively clear. Moreover, rice NAC transcription factors ONAC127/129 specifically bind the promoter of *OsSWEET4*, as revealed by ChIP-seq analysis [100]. As mentioned above, both *OsSUT1* and *OsSWEET11*, *14* are targeted by a DNA-binding protein OsDOF11 [68]. These facts indicate that a TF protein can simultaneously regulate the expressions of both *SUT* and *SWEET* genes, and these genes may be controlled by different factors. This also suggests that the regulation of sugar transporter genes is multifactorial and complex.

## 7. Concluding Remarks

In rice, both SUT and SWEET play conspicuous roles in assimilate translocation. They also participate in biotic and abiotic stress responses. Nevertheless, the physiological functions of most members of the SWEET family are not yet characterized. Moreover, the regulation mechanisms of these sugar transporter genes are only partially or beginning to be elucidated. In particular, the function of these transporters in sugar sensing [101] is an interesting field waiting to be addressed. A comprehensive understanding of their roles in planta may help to increase crop yield and quality in the future.

## Figures and Tables

**Table 1 ijms-22-11198-t001:** Rice *SUT* genes and their physiological functions.

Genes	Tissue Expression	Function/Knockout Effect	References
*OsSUT1*	Leaf; spikelet; root; endosperm; caryopsis; rachis/branch	Seed-filling; sucrose phloem loading for long-distance transport; plant growth; seed germination; pollen development	[13,22,23,24,25,26,27,28,29,30,31,32,33,34,35,36]
*OsSUT2*	Mesophyll; cross cell; lateral root; pedicel; seed; germinating seed	Seedling growth; pollen development; plant growth	[43,44,45]
*OsSUT3*	Pollen	Pollen development	[46,47]
*OsSUT4*	Leaf; root; anther; pollen; glume; embryo; caryopsis; spikelet	Reduced plant height and tiller number; yield loss of the mutants	[48,49,50]
*OsSUT5*	Culm; leaf; floret; caryopsis; embryo	Reduced seed-setting rate and increased endosperm chalk in the mutant caryopses; yield loss	[51,52]

**Table 2 ijms-22-11198-t002:** Expression levels of *SUT* genes in various rice tissues based on gene chip data (https://ricexpro.dna.affrc.go.jp/, accessed on 22 August 2021).

Gene trans-	Accession	Root	Stem	Leaf-b	Sheath	Inflore	Anther	Pistil	Le/Pa	Ovary	Embryo	Endo
*OsSUT1-2*	AK100027	1	9	7	5	0	0	1	5	7	11	3
*OsSUT1-3*	D87819	1	9	8	6	0	0	1	6	8	14	4
*OsSUT2-1*	AK067030	5	7	5	4	1	1	1	3	2	1	1
*OsSUT2-2*	AB091672	3	4	4	2	1	1	1	1	1	1	0
*OsSUT3*	AB071809	0	0	0	0	0	0	0	0	0	0	0
*OsSUT4-1*	AY137242	1	1	0	0	0	0	1	1	1	1	0
*OsSUT4-2*	AK065430	1	1	0	0	0	1	1	1	1	1	0
*OsSUT5*	AK073105	0	0	0	0	0	1	1	2	1	0	0

Red squares represent strong expression; brown squares represent medium expression; yellow squares represent weak expression; blue squares represent no or only marginal expression. Numbers in squares represent relative intensities of gene expression. Leaf-b: leaf blade; Infore: inflorescence; Le/Pa: lemma/palea; Endo: endosperm. NA: data not available. Hyphenated and numbered genes denote different transcripts.

**Table 3 ijms-22-11198-t003:** Rice SWEET genes and their physiological functions.

Genes	Expression Site	Function In/Knockout Affect	References
*OsSWEET4*	Spikelet; leaf	Seed-filling	[74]
*OsSWEET11*	Caryopsis; anther; sheath; pedicel; leaf; pollen development	Seed-filling; pollen development	[75,76,77,78,79]
*OsSWEET14*	Caryopsis	Seed-filling	[77]
*OsSWEET15*	Caryopsis; pollen; ovule vascular; leaf	Seed-filling	[76]

**Table 4 ijms-22-11198-t004:** Expression levels of *SWEET* genes in various rice tissues based on gene chip data (https://ricexpro.dna.affrc.go.jp/, accessed on 22 August 2021).

Gene trans-	Accession	Root	Stem	Leaf-b	Sheath	Inflore	Anther	Pistil	Le/Pa	Ovary	Embryo	Endo
*OsSWEET1a*	AK099531	25	13	6	14	7	14	4	14	2	10	0
*OsSWEET1b*	AK063475	0	0	7	2	0	0	0	0	0	1	4
*OsSWEET2a*	AK104255	1	0	0	0	0	0	0	0	0	0	0
*OsSWEET2b*	AK059965	0	0	1	1	0	0	0	0	0	0	0
*OsSWEET2c*	NA	NA	NA	NA	NA	NA	NA	NA	NA	NA	NA	NA
*OsSWEET3a*	NA	0	0	0	0	0	0	0	0	0	0	0
*OsSWEET3b*	NA	0	0	0	0	0	0	0	0	0	0	0
*OsSWEET4*	AK071676	6	14	8	8	7	5	6	20	7	10	2
*OsSWEET5*	AK069614	0	0	0	0	0	7	0	0	0	0	0
*OsSWEET6a*	NA	1	0	0	0	0	1	0	0	0	0	0
*OsSWEET6b*	AK099440	1	0	0	0	0	1	0	0	0	0	0
*OsSWEET7a*	NA	0	0	0	0	0	0	0	0	0	0	0
*OsSWEET7b*	NA	0	0	0	0	0	0	0	0	0	0	0
*OsSWEET7c*	NA	NA	NA	NA	NA	NA	NA	NA	NA	NA	NA	NA
*OsSWEET7e*	NA	NA	NA	NA	NA	NA	NA	NA	NA	NA	NA	NA
*OsSWEET11*	AK106127	0	3	0	3	6	10	4	12	26	0	15
*OsSWEET12*	AK109114	0	0	0	0	0	0	0	0	0	0	0
*OsSWEET13*	CI437556	3	12	6	14	0	0	0	5	3	0	0
*OsSWEET14*	AK101913	4	1	1	2	0	0	0	0	2	6	0
*OsSWEET15*	AK103266	0	4	0	1	1	2	1	6	9	4	8
*OsSWEET16*	CI149956	0	4	1	1	0	0	0	3	0	1	0

Red squares represent strong expression; brown squares represent medium expression; yellow squares represent weak expression; blue squares represent no or only marginal expression. Numbers in squares represent relative intensities of gene expression. Leaf-b: leaf blade; Infore: inflorescence; Le/Pa: lemma/palea; Endo: endosperm. NA: data not available. Hyphenated and numbered genes denote different transcripts.

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
