# Peer review of "Rice SUT and SWEET Transporters"

_ijms, 2021, doi:10.3390/ijms222011198_

Round 1

Reviewer 1 Report

This article reviews 88 papers associated with SUT and SWEET functions especially in rice. Total two tables and two figures are provided. Some recent information in this field was updated in this review article.  

  1. Common error of this review, “SUT”/ “SWEET” gene expression should be italicized in text. Some are highlighted in yellow color in the attached pdf.   
  2. Please add full name of “SE-CC “ in the first mentioned.
  3. Line60, “SUT trans-60 porters, however, cannot undertake the transport of glucose because it is not a substrate 61 for them. “ please cited reference.
  4. Line 79, “a virus SemiSWEET may be a new choice for the breakthrough in this area.” Not clear. What virus? Please revise this sentence.   
  5. Line 82-84, suggest to revise to “ To fully understand these sugar transporter’s functions are challenging, particularly in the aspect of gene regulation mechanisms.“.
  6. Line 155, please edit Table 1 and make a line to separate different genes. Please keep column to left. Table 1. Rice SUT (italicized) genes and their physiological functions. Please change “Expressional site” to “Tissue expression”. “Function/Knockout influence“ please change to “Function in/ Knockout affect”.
  7. Line 82, “physiological roles and molecular mechanism” change to “physiological roles and molecular mechanisms.”
  8. Line 152-153 revise sentence. Not clear.
  9. Line 156, “Table 2. Expressional sites and intensities of SUT genes in rice tissues based on gene Chip data “ change to “Table 2. Gene expression levels of SUT genes in various rice tissues based on gene Chip data”.   
  10. Line 189~192, revise to “… Chen et al. (2018) demonstrated that knockout of OsHAK1 induced K deficiency reduced the expression of OsSUT1, OsSUT2, OsSUT4 and OsSUT5 both at the vegetative and reproductive growth stages of rice. “

Author Response

Dear reviewers:

Thank you very much for giving us many valuable suggestions!

 Please see the attachment for detailed information.

Reviewer 2 Report

Dear Authors,

Presented review entitled ‘Rice SUT and SWEET transporters’ by Hu, Z. at all is well structured and well described sugar transporter gene’s expression profiles based on chip data and elaborate research advances. 

The introduction provides sufficient background and includes all relevant references. 

This review described physiological roles and molecular mechanisms of two classes of sugar transporters in rice. 

In my opinion review could be improved, English as well. Table 1. Rice SUT genes and their physiological functions – is not present clear. 

Author Response

Dear Reviewer,

Thank you very much for your encouraging comments and suggestions.

Please see the attachment foe detailed information. 
